# SELF-LABELING OF FULLY MEDIATING REPRESENTATIONS BY GRAPH ALIGNMENT

## ABSTRACT

To be able to predict a molecular graph structure ($W$) given a 2D image of a chemical compound ($U$) is a challenging problem in machine learning. We are interested to learn $f : U \rightarrow W$ where we have a fully mediating representation $V$ such that $f$ factors into $U \rightarrow V \rightarrow W$. However, observing V requires detailed and expensive labels. We propose **graph aligning** approach that generates rich or detailed labels given normal labels $W$. In this paper we investigate the scenario of domain adaptation from the source domain where we have access to the expensive labels $V$ to the target domain where only normal labels W are available. Focusing on the problem of predicting chemical compound graphs from 2D images the fully mediating layer is represented using the planar embedding of the chemical graph structure we are predicting. The use of a fully mediating layer implies some assumptions on the mechanism of the underlying process. However if the assumptions are correct it should allow the machine learning model to be more interpretable, generalize better and be more data efficient at training time. The empirical results show that, using only 4000 data points, we obtain up to 4x improvement of performance after domain adaptation to target domain compared to pretrained model only on the source domain. After domain adaptation, the model is even able to detect atom types that were never seen in the original source domain. Finally, on the Maybridge data set the proposed self-labeling approach reached higher performance than the current state of the art.

## 1 INTRODUCTION

Chemical compounds are often represented by a graph representation of their chemical structure. These graph representations are actually a simplification of the chemical compound as it loses some information about the electronic structure of the molecule. However, in the field of drug discovery this graph representation is often used as valuable input for machine learning pipelines. Examples of formats describing the graph representation of a chemical compounds are SMILES [36] and MOLfile [5]. However, especially in patents but also in scientific literature the chemical compound is only described using an image format. Automatically recognizing the chemical structures on these images is valuable for machine learning approaches to be able to process these sources of chemical compounds.

Learning to recognize a graph structure from 2D images of chemical compounds seems like a fairly simple task for humans. However, for machine learning models it seems that generalization to new domains of images (e.g. different line width, font face) [21] is not happening naturally. When we humans see an image with a graph structure that we do not recognize completely, we start reasoning and analyzing the part of the graph we are not sure about. We humans automatically align the graph part we recognized on the image with the complete graph including the unrecognized part of the graph. One way to finish our graph prediction is to guess the unknown nodes or edges after which we check for correctness. If the graph prediction was correct we know that this guess was most probably correct and we could try to apply this new knowledge to other images.

To be able to do this reasoning on for example images using graph alignment in machine learning we need a detailed (on pixel level) representation. Therefore we assume a fully mediated model [2] where we are interested to learn $f : U \rightarrow W$ having a fully mediating representation $V$ such that $f$ factors into $U \rightarrow V \rightarrow W$, which is visualized in Figure 1. Thus, in order to predict

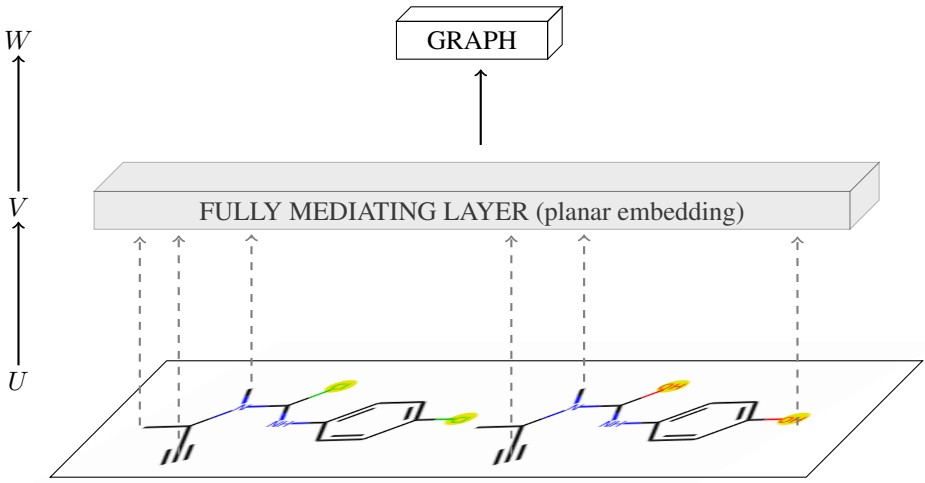

Figure 1: We are interested to learn $f : U \to W$ having a fully mediating representation $V$ such that $f$ factors into $U \to V \to W$. In the case of optical graph recognition of chemical compounds from 2D images, the fully mediating layer is represented using the planar embedding of the chemical graph structure we are predicting.

$W$ from $U$ we first need to pass the fully mediating layer, no side paths are allowed. When a fully mediating representation is used some assumptions [23; 26; 25] are made about the mechanism of the underlying process. This mechanistic prior restricts the space of possible models to all the models that follow the mechanistic assumption. We hypothesize that the use of this richer representation (fully mediating representation) enables for a better generalization. Additionally, as an interesting side effect, we observe that the mechanistic assumption allows for a better interpretability of the underlying model.

In the case of optical graph recognition of chemical compounds from 2D images, the fully mediating layer is represented using the planar embedding of the chemical graph structure we are predicting. In order to learn the planar embedding of a chemical graph structure, we start from a model described in Oldenhof et al. [21] which has two steps: an image segmentation and an image classification step. To train this model, **pixel-wise** labels are needed for every image describing precise locations of nodes and edges in the graph (planar embedding) which we will call rich or detailed labels in our setup. However, these rich labels are not always available and implies a manual process where intermediate organic chemistry knowledge is required. In the more common cases, data sets only contain 2D images of chemical compounds ($U$ in Figure 1) and on the other side the final output in SMILES[36] or MOLfile[5] format ($W$ in Figure 1). These formats describe the graph structure of the chemical compound but not the particular planar embedding of this graph structure ($V$ in Figure 1) in the context of the image. To solve this problem, we propose a **graph aligning** approach that generates rich labels $V$ given normal labels $W$. This method would enable learning of the fully mediating representations given only normal labels $W$.

In section 4 we empirically evaluate our domain adaption method. We observe that compared to the non-adapted model we drastically increase accuracy even on atoms and bond that were not present in source domain.

*Key contributions*: (1) we propose a novel rich labeling framework by introducing the use of fully mediating representations, (2) in the case of graph recognition we show that the rich labeling can be performed by graph alignment, (3) we show it enables data efficient domain adaption and (4) reaches state-of-the-art performance on Maybridge compound data set.

## 2 RELATED WORK

**Structural scene representation and visual reasoning.** Our work has similarities with research done on structural scene representation and visual reasoning [11; 41; 19]. The disentanglement of the reasoning and the representation described in Yi et al. [41] enables the model to solve complex reasoning tasks. In our work the complex reasoning task would be graph alignment which is disentangled from the optical graph recognition.

**Slot Attention.** Our method is related with a method called slot attention [17] where the Hungarian algorithm [16] is incorporated in a model for object detection. This Hungarian algorithm is limited to only sets while in our case we need to map more complicated structures composed of different atoms connected with different bond for which we need graph alignment in order to adapt iteratively a model to a new target domain.

**Image to Graph methods.** In the field of computational chemistry there are several tools available [22; 34; 20; 7; 21; 18; 33] to convert an 2D image of a chemical compound to a SMILES [36] format or similar which in fact represents a graph structure of a chemical compound. Also for road extraction from satellite images there are several methods available [3; 8; 12]

**Graph Matching.** In computer vision graph alignment is usually known as graph matching. It can be useful to (1) locate objects from features [10], (2) to transfer knowledge [42] and (3) to find matches in database [13]. Also for comparing social networks graph matching can be very important to allow to uncover identities of communities [14]. In chemistry, comparing graphs can be helpful to identify identical chemicals, substructures or maximum common part of chemicals. In the work of Willett et al. [37] an overview is presented about the use of similarity searches in chemical databases.

**Domain Adaptation** In the work of Kouw & Loog [15] a comprehensive overview is given for domain adaptation methods when labels for the target domain are not available. Our method has some similarities with semi-supervised iterative self-labeling [4; 27] approaches where predictions on a data set of a new domain of a pre-trained model are used as pseudo-labels and used to retrain the model again iteratively until convergence. In the work of Das & Lee [6] even a graph matching loss is first used to learn a domain invariant representation for source and target domain after which the use of pseudo-labels show a significant improvement of performance. In our work the graph matching is used for a different purpose as opposed to the the work of Das & Lee [6]. Graph matching is used in our work to generate rich labels given the 'normal' labels we have from the target domain. This is where our method also differs from other semi-supervised methods for domain adaptation when no target label information at all is assumed and no distinction is made between rich and 'normal' labels.

**Weak Supervision** In our setup we use the term 'rich' or 'detailed' labels to differentiate from the normal labels. We would like to contrast these 'rich' labels with the term 'strong labels' used in the setting of weak supervision. For example, in the machine learning task of image segmentation pixel-wise labels are needed which are expensive and often not readily available. Therefore, weak supervision methods have been developed to address this issue. Weak supervision can be used to help image segmentation by only using image labels (no pixel-wise labels) [35; 38]. A more general framework was presented in Xu et al. [39] to be able to learn semantic segmentation from a variety of types of weak labels (*e.g.*, image tags, bounding boxes and partial labels). Another approach is to augment the strong labeled data set using weakly labeled data [40]. However, the main difference with all of the methods mentioned above is that our method does not work on weak labels because our end goal is different. The main goal of our machine learning approach is to help to predict 'normal' labels by using rich labels.

**Front Door Criterion** Our framework exploits fully mediating variables. A variable is called a mediator when it meets several conditions regarding the relationship with other variables as described in Baron & Kenny [2]. Another perspective of the mediating relationship is given by Pearl [23]; Pearl et al. [26]; Pearl [25] who introduce the front door criterion where the mediator actually enables to estimate unbiased causal effects. A more formal interpretation of these causal effects is presented in Pearl [24]. In order to use a mediating model the mediator needs to be identified or assumed first, which is not always straightforward. In our setup (see Figure 1), the assumption means that the relation between input $u \in U$ and planar embedding $v \in V$ is a map and as well as the relation between $v$ and the final graph $w \in W$. Furthermore, we assume no side paths from $u$ to $w$.

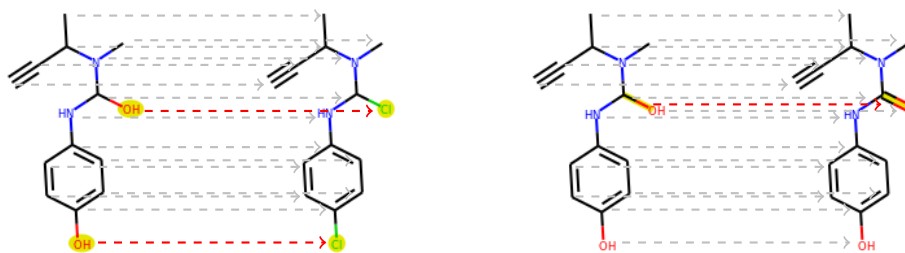

(a) Example of 2 chemical compounds with graph edit distance of 2 node substitutions.

(b) Example of 2 chemical compound with graph edit distance of 1 edge substitution.

Figure 2: Two examples of chemical compounds graphs with their graph edit distance. The nodes of the graphs are first aligned before computing the graph edit distance. The node alignments are marked with the gray dashed arrows. The differences after graph alignment are highlighted and the substitutions are marked with the red dashed arrows.

## 3 SELF-LABELING OF FULLY MEDIATING REPRESENTATIONS

Our goal is to learn $f : U \to W$ assuming a fully mediating representation $V$ such that $f$ factors into $U \to V \to W$. In order to learn the first part of $f$ ($U \to V$) we need labels for $V$ which are expensive in the case of optical graph recognition of chemical compounds from 2D images where $V$ is represented as the planar embedding of the chemical graph structure. Our method tries to address this issue by iteratively updating the model using self-labeled labels for $V$ by graph aligning the graph predictions using the model from previous iteration with the given true graphs (labels $W$).

### 3.1 GRAPH ALIGNMENT

A possible and often used closeness score to compare graphs is the **graph edit distance** [29]: given 2 graphs, not necessarily of equal size and a set of operations, that are $\mathcal{O} = \{\text{vertex/edge/label insertion/deletion/substitution}\}$, and a cost function $c : \mathcal{O} \mapsto \mathbb{R}$, so we find the cheapest sequence of operations that convert $\mathcal{G}_1$ into $\mathcal{G}_2$, which translates to an optimization problem:

$$\min_{\{e_i\}_{i=1}^k \in \mathcal{O}^k : \mathcal{G}_2 = (e_k \circ \ldots \circ e_1) \times \mathcal{G}_1} \sum_{i=1}^k c(e_k),$$

Although there are some efficient algorithms available [30; 31; 32] in order to compute the graph edit distance, it remains a computational hard problem.

Closely related with the concept of graph edit distance we introduce for our method the map $E(v)$ which gives the allowed operations on a given graph $v$ given a specific constraint. This constraint is a parameter which can be tuned for a specific data set or problem domain. Examples of such constraints are maximum 2 node substitutions or maximum 1 edge substitution as shown in Figure 2.

### 3.2 METHOD

Let us now say we have a trained neural network model for $f : U \to V$, a projection $\phi : V \to W$, a pair $(u, w)$ of input $u \in U$ and normal label $w \in W$ and we would like to infer rich label $v \in V$ from the given datapoint $(u, w)$. We also assume the map $E(v)$ which gives all allowed graph edits for the graph $v$. Let $\hat{v} = f(u)$ be the predicted rich label from the model, then we define a term correcting edit as

**Definition 3.1** *Edit $e$ is a **correcting edit** if when $e$ is applied to the prediction $\hat{v}$ and then projected to the $W$ space the resulting graph is the true graph $w$ (up to isomorphism),* i.e.,

$$\phi(e \times \hat{v}) \cong w,$$

*where $\times$ is the application of edit to the planar embedded graph $\hat{v}$.*

Notice that for a given $\hat{v}$ and $w$ there can be multiple edits that are **correcting edits** which create a dilemma of choosing the best **correcting edit**. Therefore, we make the following assumption:

**Assumption 3.1** *The probability that a correcting edit $e$ results in the true underlying rich label $v$ is monotonely decreasing with respect to the size of edit $e$ (*i.e., $|e|$*).*

*In other words, if we take two correcting edits $e_1, e_2$ then we assume the following:*

$$|e_1| < |e_2| \quad \Rightarrow \quad P(e_1 \times \hat{v} = v) > P(e_2 \times \hat{v} = v)$$

The assumption is based on the fact the *probability of any individual mistake in a graph* by the model is low. This is because if the probability of a mistake would be high the model would not be able to produce a graph with a total of 1-2 edit distance. Thus, the graphs with few edits have low mistake probability and for them the Assumption 3.1 is valid.

Then we use the following optimisation problem to find the best correcting edit $e$ to convert $\hat{v}$ to rich label $v$ for input $u$:

$$\mathcal{E}^* = \arg\min_{e \in E(\hat{v})} |e| \quad \text{such that} \quad \phi(e \times \hat{v}) \cong w,$$

where $\arg\min$ returns the set of minimal solutions or the empty set if no solutions exist.

There are three possible outcomes of last mentioned optimization problem: (1) no solution is found, (2) a single $e$ is found or (3) multiple equal size $e$ are found. In the optimal case (2) a single $e$ is found so we can label a new $v$ for our given datapoint $(u, v)$. In the case of (1) when no solution is found, no new $v$ is labeled. In the last case (3) when multiple equal size solutions are found there are four options we could do. First (3.1), we could discard the solutions and not label $u$. Second (3.2), we could take $e$ that results in the highest likelihood for $e \times \hat{v}$ based on the model $f$. Third (3.3), a solution $e$ is picked uniformly randomly in order to generate the rich label label $v$. Fourth (3.4), pick $e$ randomly according to the likelihood of $e \times \hat{v}$ in the model $f$.

This process is repeated for every datapoint $(u, w)$ we have available from the target domain. Thus, several new labels $v$ are found for different datapoints. Once all datapoints are processed these new rich labeled datapoints are added to the training data set after upsampling and our model can be retrained. After this, a new iteration begins and all available datapoints $(u, v)$ are again processed to find even more new rich labels $v$ and we can retrain the model again. This iterative process can be repeated until convergence (see Algorithm 1).

---

**Algorithm 1:** Iterative algorithm for Self-Labeling of Fully Mediating Representations

**Data:**
Target domain data $\mathbf{L} = \{(u_i^t, w_i^t)\}_{i=1}^n$
Source domain data $\mathbf{S} = \{(u_j^s, v_j^s)\}_{j=1}^m$ (rich labels)
**Result:** $f : U \to V$
**repeat**
    // Inferring rich labels for target data
    $\mathbf{T} = []$;
    **for** $(u, w)$ *in* $\mathbf{L}$ **do**
        $\hat{v} \leftarrow f(u)$;
        $\mathcal{E}^* \leftarrow \arg\min_{e \in E(\hat{v})} |e| \quad \text{such that} \quad \phi(e \times \hat{v}) \cong w$;
        **if** $\mathcal{E}^*$ *is a not empty* **then**
            $e \leftarrow \text{choose}(\mathcal{E}^*)$;
            $v \leftarrow e \times \hat{v}$;
            $\text{appendRichLabels}(\mathbf{T}, (u, v))$;
        **end**
    **end**
    $\mathbf{T} \leftarrow \text{UpSample}(\mathbf{T})$;
    $f \leftarrow \text{RetrainModel}(\mathbf{S}, \mathbf{T})$;
**until** $\text{Converged}(f)$;

---

| Dataset | Orig. Size | Tot. #samples | Cand. rich labeling #samples | Test #samples |
|---------|-----------|---------------|------------------------------|---------------|
| Indigo | 50,000 | 5,000 | 4,000 | 1,000 |
| Maybridge | 5,740 | 5,000 | 4,000 | 1,000 |

Table 1: Summary of datasets from the 2 different target domains

## 4 EXPERIMENTS

For the experiments we focus on the problem of predicting chemical compound graphs from 2D images where the fully mediating layer is represented using the planar embedding of the chemical graph structure we are predicting. In order to measure empirically the performance of our method of self-labeling fully mediating representations we perform three steps. (1) We pre-train (training details in Appendix A.2) a ChemGrapher [21] model (summarized in Appendix A.1) wherefore, corresponding to the pipeline described in the work of Oldenhof et al. [21], we sample around 130K chemical compounds from ChEMBL [9] in SMILES format and artificially generate, using an RDKit fork [1], a rich labeled dataset with 2D images of chemical compounds. (2) Secondly, we test the baseline performance of this pre-trained model on two different test sets from two different target domains than the source domain of the pre-trained model. (3) Thirdly, we apply our domain adaptation method and measure performance again on the two target domains.

For the first target domain we take a data set from the work from Staker et al. [33], which we will call Indigo data set. For the second target domain we take the data set which was published by the developers of MolRec [18] which we will call the Maybridge data set. Both data sets provide 2D images from a chemical compound together with corresponding identifier of a the chemical compound like SMILES [36] or MOLfile [5]. These identifiers describe the graph structure of the chemical compound however they do not provide the planar embedding of the graph (e.g. no information about the pixel coordinates of every node or edge in the image). Visually we can also observe that the Maybridge dataset contains images where the style is closer related to the training images style used for the pre-trained model compared with the images in Indigo dataset where the style of images is quite different. Therefore we expect a significant worse starting performance of the pre-trained model on the Indigo dataset compared with the Maybridge dataset.

From both data sets we randomly sample 5,000 datapoints which are split in 4,000 datapoints used for our method and 1,000 datapoints to measure performance on (summarized in Table 1). When processing the 4,000 datapoints our method will be able to generate rich labels for the datapoints where the graph prediction could be graph aligned with the true graph. As the number of rich labeled datapoints this way is maximum 4,000 we will upsample them (x number of copies) before adding them to the training data set. In our experiments we differentiate between two strategies of upsampling. One way is to upsample all the rich labeled data points equally from the target domain to a fixed number, for example 20,000. Another way is to take into account, while upsampling, the number of atom types that are rich labeled and make sure that the rare atom types are upsampled to a specific threshold.

One important tuning parameter in our method is the number of allowed operations. For our experiments we will try two different values for this parameter. Firstly, we set this parameter to zero meaning we do not allow any operation for graph alignment. We will call this **exact graph alignment**. Secondly, we allow a maximum of 2 node substitutions or a maximum of 1 edge substitution for graph alignment, which we will call **correcting graph alignment**.

In total we will measure the performance of 4 variations of our method (varying allowed operations and upsampling strategy) on both data sets. The performance we will measure is the accuracy of $U \rightarrow W$ as we only have access to the normal labels of target domain. However, we assume that if the final graph prediction is correct ($W$) it is highly likely that also the planar embedding ($V$) is correct. As our method is an iterative method we will report results for every iteration starting with the initial performance before applying our method. The results of these experiments are summarized in Figure 3. We observe that all variations of our method are able to improve performance on target domain compared with initial pre-trained model on source domain. On the Indigo data set the best variation is even able to obtain 4x improvement. The best variation of our method on the Indigo data set was using **correcting graph alignment** without upsampling of rare atom types

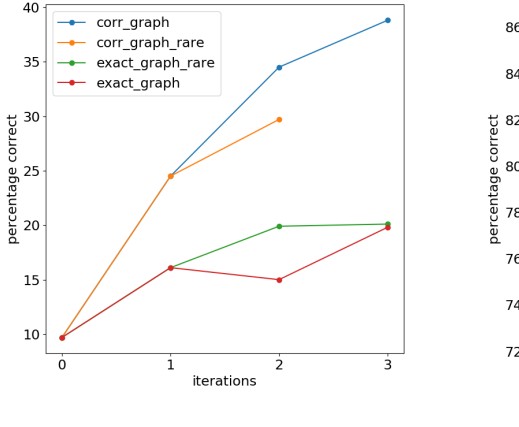
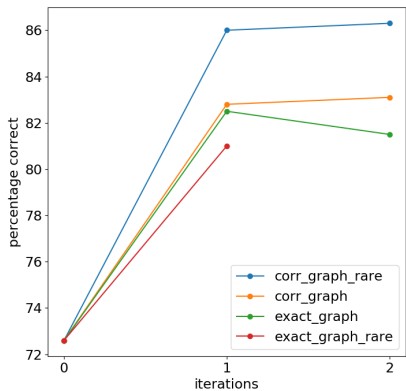

(a) Results on Indigo data set        (b) Results on Maybridge data set

Figure 3: Comparison performance of methods on Indigo and Maybridge data set. Self-labeling by **correcting graph alignment** is clearly better performing than when **exact graph alignment** is used. Sometimes upsampling of rare atoms to a specific threshold (note postfix _rare_) before retraining of model can boost performance. Performance on target domain at iteration 0 is the performance of pre-trained (on source domain) ChemGrapher before domain adaptation.

while on the Maybridge data set the best variation was also using **correcting graph alignment** but with upsampling of rare atom types. Some of the underperforming variations of our method were stopped early in order to save computational resources.

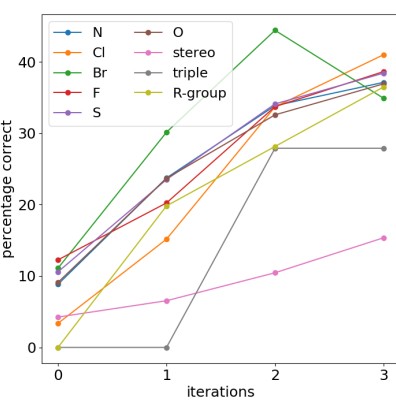
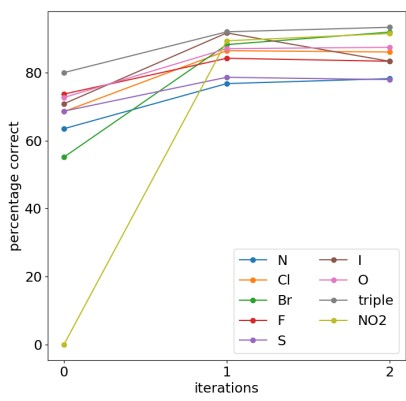

(a) Results on Indigo data set        (b) Results on Maybridge dataset

Figure 4: We take the best performing methods and analyze their performance on different atom and bond types per iteration. We observe that for some atom types the method is able to increase performance even though initial performance was 0%. This is the case in for example R-groups in Indigo data set or superatom $NO_2$ in Maybridge data set.

We choose the best variation of our method for every data set and analyze the performance on different atom and bond types per iteration. We measure for every atom or bond type the percentage of graphs predicted correctly from the total number of graphs containing that specific atom or bond type per iteration, which is visualized in Figure 4. Most of the performances of the different atom and bond types increase per iteration for both data sets even when initial performance was 0%.

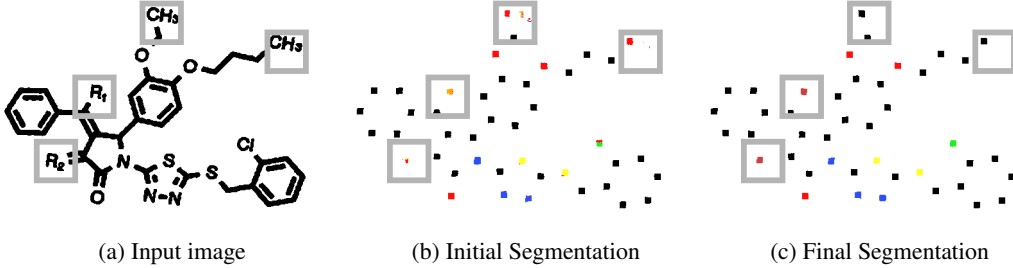

|(a) Input image|(b) Initial Segmentation|(c) Final Segmentation|

Figure 5: Comparison initial segmentation with final segmentation after applying self-labeling of fully mediating representations for Indigo data set. We observe that the initial model is making mistakes on the R-group atom type and carbon represented with a 'C'. In the final model we see that now predictions are all correct.

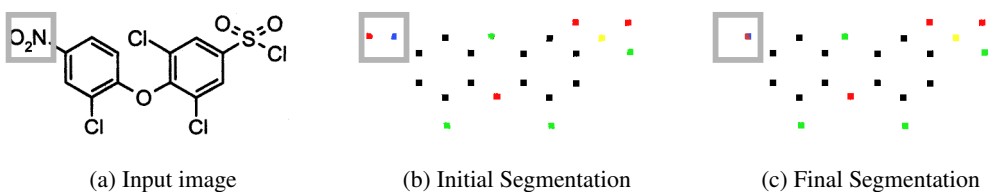

|(a) Input image|(b) Initial Segmentation|(c) Final Segmentation|

Figure 6: Comparison initial segmentation with final segmentation after applying self-labeling of fully mediating representations for Maybridge data set. The initial model predicts the superatom $NO_2$ as two separate atoms O and N which is chemically not correct. The final model makes the correct prediction.

The atom types where initial performance was 0% are atom types never seen before in source domain. For example in the Indigo data set there are compounds with atom labels like R1, R2 and R3 representing R-groups which were not present in the original data set from the source domain. For illustration purposes we visualize in Figure 5 the segmentation step which forms part of the graph recognition model used in this study. In the initial segmentation from the pre-trained model we can clearly see that the model confuses the R-group atoms with the oxygen atom type and the hydrogen atom type. After applying our method the model is able to make correct predictions. In the same Figure 5 we also observe that in the Indigo data set carbon sometimes also is represented using a C which was never the case in the original data set. The initial segmentation mainly confuses these carbon atom types with the oxygen atom type. After applying our method the model again makes the correct prediction.

Similarly, the superatom $NO_2$ present in the Maybridge data set was never observed in the source domain. However, again after applying our method the model is able to detect superatom $NO_2$ correctly. We illustrate the segmentation step of the graph recognition model in Figure 6 for an example image taken from the Maybridge data set. We observe that in the initial segmentation the pre-trained model confuses $NO_2$ with nitrogen atom and also oxygen atom which chemically is not the correct prediction. In the final segmentation after applying two iterations of our method the newly trained model is able to make the correct prediction.

Additionally Figure 5 and Figure 6 also show an interesting side effect when using a fully mediating representation. Consider a classical model where input is an image and output is SMILES. When the output prediction of the model is incorrect it is not clear in which part of the image the mistake was made but in the case of having available the planar embedding (mediation representation) the expert can see where and how the mistakes happened. This makes the model more interpretable.

Finally we compare in Table 2 the resulting best performance of the model after applying our method on the Maybridge data set with several other methods available. We observe that our approach enables to reach higher performance than the current state of the art. For the freely available tools OSRA [7] and Molvec [28] we measured the performance using the same randomly 1000 datapoints from the Maybridge dataset. For MolRec [18] this was not possible but we report for information

| Method | Training Dataset | | Maybridge Test Dataset | Accuracy |
|---|---|---|---|---|
| | Source domain | Target domain | | |
| OSRA (v2.1.0) [7] | N/A | N/A | same random 1,000 | 80.4% |
| Molvec (v0.9.8) [28] | N/A | N/A | same random 1,000 | 78.4% |
| ChemGrapher [21] | 130K images | N/A | same random 1,000 | 72.6% |
| ChemGrapher [21] (using manually rich-labeling) | 130K images | 40 manually handpicked and rich-labeled images (upsampled) | same random 1,000 | 81.6% |
| **Proposed domain adaptation** | 130K images | 4,000 non-rich labeled | same random 1,000 | **86.3%** |
| MolRec [18] | N/A | N/A | Total 5740 | 83.8%[from [18]] |

Table 2: Comparison performance on Maybridge data set. We observe that our approach enables to reach higher performance than the current state of the art. Most of the tools available for chemical graph recognition are rule based approaches for which a training dataset is not relevant.

the performance on the total Maybride dataset as reported in the work of M. Sadawi et al. [18]. Finally for ChemGrapher [21] we measured performance using three different training datasets. Firstly, we measure the performance when we only have access to the source domain (generated using RDKit [1]). Secondly, we measure performance using the same training dataset from source domain but adding upsampled (100 copies) 20 handpicked manually rich labeled datapoints from the target Maybridge domain (as was done in the work of Oldenhof et al. [21]). Finally, instead of manually rich labeling datapoints, we process the 4,000 datapoints from Maybridge target domain where our method will be able to generate rich labels for the datapoints where the graph prediction could be graph aligned with the true graph, after which these rich labeled datapoints are added to the training dataset.

## 5 CONCLUSION

Machine learning models often are faced with the problem to not generalize well to a new domain. This is also the case for chemical graph recognition from images. We have shown that fully mediating layers can be exploited in machine learning models to adapt in data efficient way to new domains, without the need of rich expensive labels as they can be generated using our method. In the case of chemical graph recognition we empirically show that our method is able to adapt to a new domain of chemical compounds, with **previously unseen** atom or bond types. Our rich-labeling method required only 4,000 normal labeled points in the target domain to go from 10% accuracy to 39%, i.e., almost 4x improvement in the difficult Indigo data set. Furthermore, on Maybridge data set, again using only 4,000 images, we reached high accuracy obtaining better performance than the current state of the art.

Effective tools of chemical structure recognition from images enable access to the knowledge in chemical literature which is currently only available through expensive chemistry databases. We believe it as an important step towards open pharmaceutical science.

It would be interesting to apply this method to other contexts where the output of a machine learning model could be represented with a graph structure. For example, the case of structural scene representation, where a scene could be represented using a graph where every vertex could represent an object and every edge would represent the relations between the objects (e.g. side-by-side, on-top-of, under). This structural scene could be in form of 2D images or it could be even generalized to 3D models, where point clouds are available and one is interested to transform them into 3D graphs of connected parts.

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

## A APPENDIX

### A.1 ARCHITECTURE SUMMARY OF GRAPH RECOGNITION TOOL

Every iteration of our method we need to train the graph recognition tool described in Oldenhof et al. [21]. This graph recognition tool is built using a combination of different convolutional neural networks. The first part is a semantic segmentation network to pixel-wise predict every atom, bond and charge type. The second part consists of three classification networks to classify every segment predicted by the semantic segmentation network. After the first step of the ChemGrapher model [21], the segmentation network, the predicted segments are processed so that for every segment the center of mass is calculated. These centers of mass would be the atom/bond/charge candidates to be classified by the classification networks.

Table 3: Summary of the layers of the segmentation network

| Layer | Kernel | Nonlinearity | Padding | Dilation |
|-------|--------|--------------|---------|----------|
| conv1 | 3x3 | ReLU | 1 | no dilation |
| conv2 | 3x3 | ReLU | 2 | 2 |
| conv3 | 3x3 | ReLU | 4 | 4 |
| conv4 | 3x3 | ReLU | 8 | 8 |
| conv5 | 3x3 | ReLU | 8 | 8 |
| conv6 | 3x3 | ReLU | 4 | 4 |
| conv7 | 3x3 | ReLU | 2 | 2 |
| conv8 | 3x3 | ReLU | 1 | no dilation |
| last | 1x1 | none | no padding | no dilation |

Table 4: Different layers in the classification network

| Layer | Kernel | Nonlinearity | Padding | Dilation |
|---|---|---|---|---|
| depthconv1 | 3x3 | ReLU | 1 | no dilation |
| conv2 | 3x3 | ReLU | 2 | 2 |
| conv3 | 3x3 | ReLU | 4 | 4 |
| conv4 | 3x3 | ReLU | 8 | 8 |
| conv5 | 3x3 | ReLU | 1 | no dilation |
| global maxpool | input size | None | no padding | no dilation |
| last | 1x1 | None | no padding | no dilation |

A.2 TRAINING DETAILS FOR GRAPH RECOGNITION TOOL

Training details of the graph recognition tool for every iteration of our method are summarized in Table 5. The input images used for training of the different networks are a mix if images from source domain and upsampled rich labeled images from target domain. For pretraining of the ChemGrapher model only images from source domain were used. The training was performed using a compute node with 2 NVIDIA v100 GPUs with 32GB of memory.

Table 5: Training details for different networks

| Network | #input images source domain | #input images target domain (upsampled) | #epochs | walltime | minibatch size | learning rate |
|---|---|---|---|---|---|---|
| Segm. network | 114K | 20K | 5 | 24h | 8 | 0.001 |
| Atom Clas. | 12.4K | 2.6K | 2 | 8h | 16 | 0.001 |
| Charge Clas. | 12.4K | 2.6K | 2 | 8h | 16 | 0.001 |
| Bond Clas. | 4.4K | 2.1K | 2 | 4h | 64 | 0.001 |

A.3 COMPUTATIONAL COST PER RICH-LABELING ITERATION

In the following Table 6 the computational cost for 1 rich-labeling iteration is summarized including all steps: (re)training, predicting and graph aligning rich-labeling.

Table 6: Computational costs per rich-labeling iteration

| | Training | Predict | Graph Aligning | |
|---|---|---|---|---|
| Hardware | 2 NVIDIA v100 GPUs | 1 NVIDIA v100 GPU | Intel Xeon Gold 6240  2.6Ghz | |
| Dataset | Source+Target domain | Indigo/Maybride | Indigo | Maybridge |
| #datapoints | see Table 5 | 4,000 | 4,000 | 4,000 |
| Walltime | ~44h (details Table 5) | ~2h | ~40min | ~3min |

A.4 EXAMPLES OF CASES WHERE GRAPH ALIGNMENT FAILS

We would like to showcase some examples where the constrained (max 2 node substitutions or max 1 edge substitution) graph alignment fails. At the same time it is important to note that our proposed domain adaptation method is an iterative method, so if a graph alignment fails in a previous iteration it could succeed in a next one when the new model makes a new graph prediction closer to the true graph.

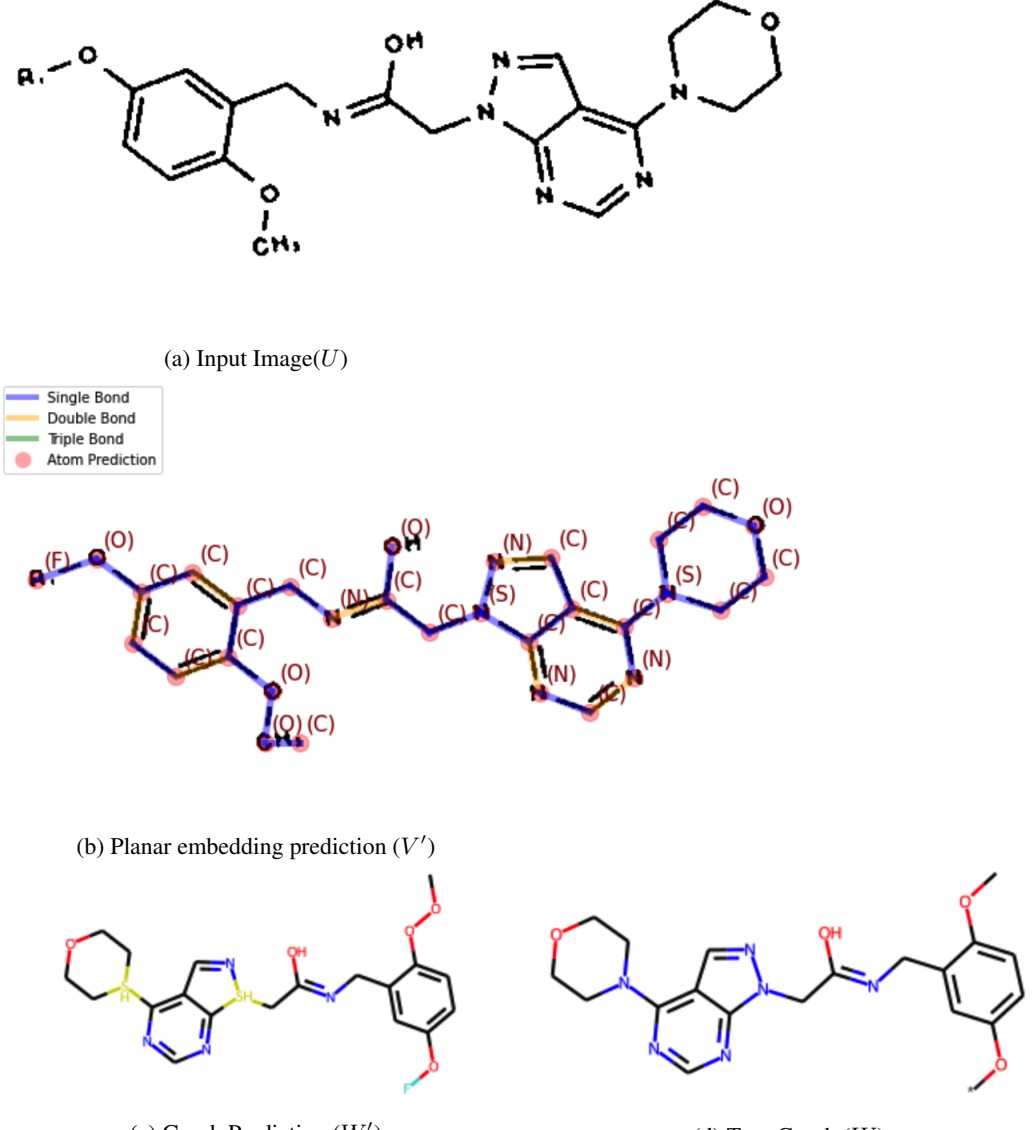

(a) Input Image($U$)

(b) Planar embedding prediction ($V'$)

(c) Graph Prediction ($W'$)

(d) True Graph ($W$)

Figure 7: Example 1: It is clear that to align the graph prediction $W'$ with the true graph $W$ more than 2 node substitutions are needed. So no rich labeling is possible for this example in this iteration.

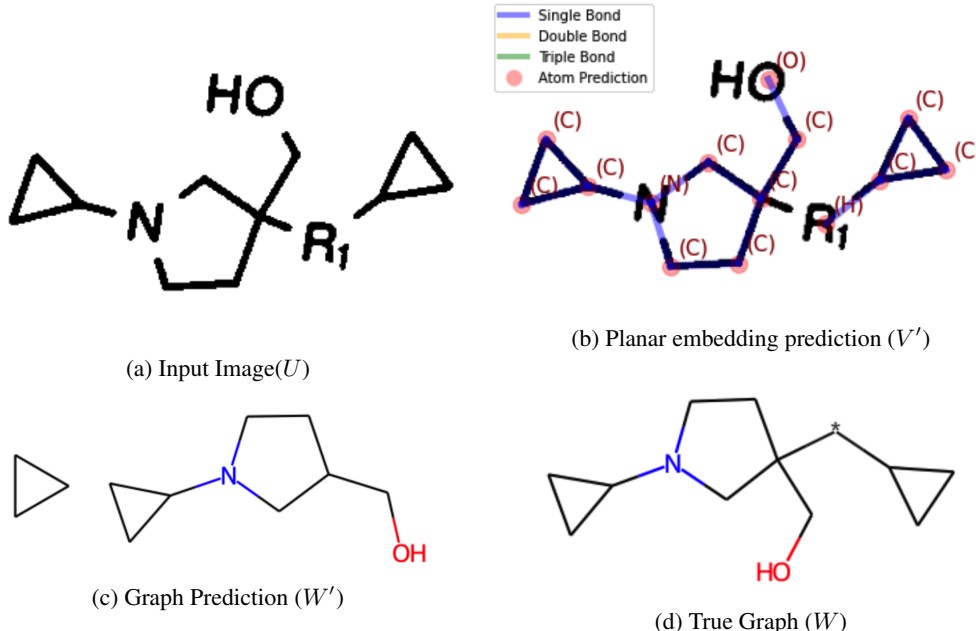

(a) Input Image($U$)

(b) Planar embedding prediction ($V'$)

(c) Graph Prediction ($W'$)

(d) True Graph ($W$)

Figure 8: Example 2: It is clear that alignment of the graph prediction $W'$ with the true graph $W$ can not be solved with only substitutions.

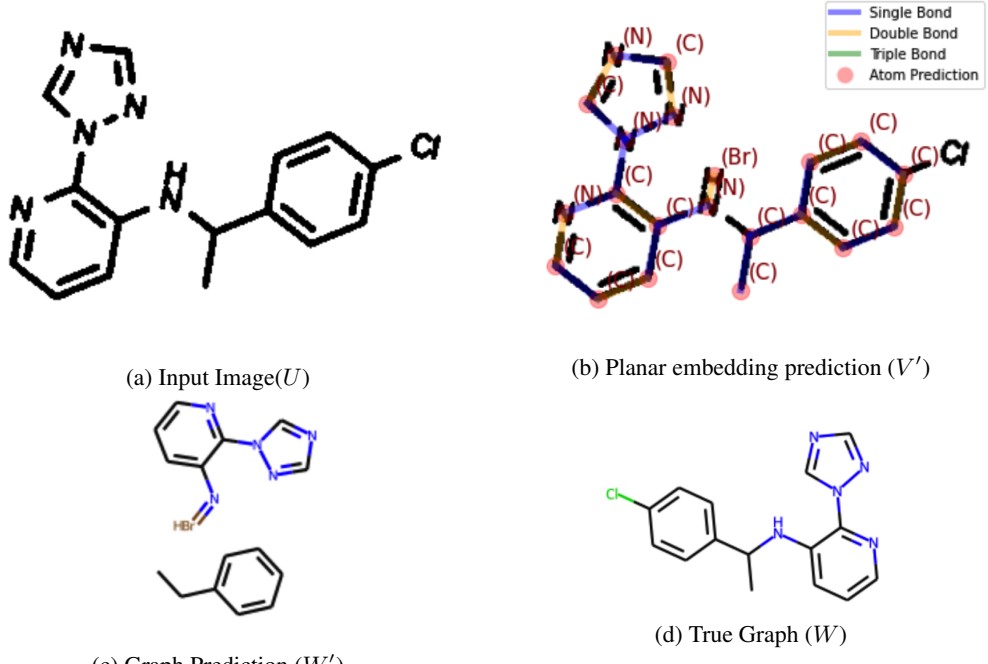

(a) Input Image($U$)

(b) Planar embedding prediction ($V'$)

(c) Graph Prediction ($W'$)

(d) True Graph ($W$)

Figure 9: Example 3: It is clear that alignment of the graph prediction $W'$ with the true graph $W$ can not be solved with only substitutions.

