# OpenReview forum: "Self-Labeling of Fully Mediating Representations by Graph Alignment"
_ICLR.cc/2021/Conference — Reject_

### Official Review · AnonReviewer2 · 2020-10-27
**Goal of the paper is to learn a function that maps an input (that can represent a graph on a 2D image) to the graph structure. Paper has experiments showing importance of mediating layer but doesn't make similar comparisions to previous work making it harder to understand if the mediating layer is really useful or not.**

**Rating:** 4
**Confidence:** 4

**Review:**

Clarity
The paper is well written. The relevant work, introduction and main text allows the user to understand the problem easily.

Originality
Paper uses intermediate layers (called mediating layers) as a concept to solve the problem of mapping  an input (that can represent a graph on a 2D image) to the graph structure. The base system for segmentation and classification is derived from previous work. The paper introduces graph aligning the mediating layer to the ground truth labels and propose an algorithm for the process. The originality is in the proposal of graph alignment as an intermediate step to generate additional training data while allowing different edit corrections.

Quality
The experimentation showing the effectiveness of using a mediating layer compared to the base seems adequate for the 2 datasets. However there doesn't seem to be a good comparison to previous work. Moreover the previous work had different sampling (number of training and test samples) and is hard to understand if Table 1 is useful i.e. the comparison of numbers aren't useful. Having a clear fair comparison would have suggested the mediating layer's use more strongly.

Significance
The work claims to improve previous performance by a few percent points but as noted above the comparison may not be proper.

Questions
1. Since the values of V are not really known (both during training and testing), why are the labels termed strong labels?
i.e. is the correctness of the planar graph verified at any stage?

2. There is a statement that the mediating layer can make the model more interpretable. But there is no further discussion in the paper on how this is true.

3. It is not clear how the system is able to correctly classify never observed atoms and bonds in test examples. Is this valid only during training and alignment? Otherwise how does it learn to classify them?

4. Above the key contribution section on page 2, there is mention that after domain adaptation, the performance is checked on the same test data. Why is the use of doing this? And why would we expect better results than not using domain adaptation?

5. How is set of V initialized? Is it an empty set in the beginning? In algo 1, input seems to mention there are already m strong labels before the start, does that need some correction? Also, perhaps one line needs update of S
S <- appendStrongLabels(T, (u,v));

6. Do iteration 0 always correspond to the Oldenhof model or were they different models, not clear from the text other than mention that it was pre-trained?




Goal of the paper is to learn a function that maps an input (that can represent a graph on a 2D image) to the graph structure. Paper has experiments showing importance of mediating layer but doesn't make similar comparisions to previous work making it harder to understand if the mediating layer is really useful or not.

---

> ### Author Response · Authors · 2020-11-17
> **Response to AnonReviewer2**
>
> Thank you for your feedback. Our plan is to update our manuscript before rebuttal deadline so we provide a comparison with previous work which is more clear.
>
> In the meanwhile we would like to respond to the questions raised:
>
> 1. Good point, we reconsidered and to avoid confusion with weak supervision we will change the term to "rich label" in the text. The correctness of the planar graph was not verified at any stage. In the domain adaptation setting we consider we assume there are no rich labels available. Thus, it is not possible to directly check the correctness of the planar graph.
> 2. Consider a classical model where input is an image and output is SMILES. When the graph prediction is incorrect it is not clear in which part of the image the mistake was made but in the case of having available the planar embedding (mediation representation) the expert can see where and how the mistakes happened. So as a side-effect the rich label provides interpretability. To showcase the interpretability we also made available a notebook on github: https://github.com/iclr2021-paper1739/workflow-paper1739 .
> 3. This was not well explained in the paper and we made now more clear. In detail, the never observed atoms and bonds refers to the source domain data for which we have rich labels. However, these atoms and bonds are present in the target domain, for which we do not have any rich labels.
> 4. The explanation was not clear, we have updated our manuscript.
> 5. V is initialised by the rich labels from the source domain, i.e., these are our generated images by RDKit. For this generation we use the code from ChemGrapher (Oldenhof et al, 2020) available in their github. Regarding the rich labels S and T: Our explanation was not clear we improved the algo so we keep T separate from S because we want to upsample T before retraining the model.
> 6. Yes, iteration 0 is ChemGrapher model trained as described in Oldenhof et al. We also added some clarification accordingly in the manuscript.

---

### Official Review · AnonReviewer1 · 2020-10-28
**Report on "Self-Labeling of Fully Mediating Representations by Graph Alignment"**

**Rating:** 5
**Confidence:** 4

**Review:**

##########################################################################
Summary:
This article presents a methodology to generate the complete molecular graph (including connectivity between atoms and functional groups) from a 2D image by using an intermediate representation (called fully mediating representation, V). This problem is of high relevance and is likely to have a considerable impact in the chemistry community. The particular advance of this work is to include such intermediate representation based on a graph alignment approach that generates “strong” labels.
##########################################################################
Reasons for score:
Overall, I vote for not accepting. As mentioned before, this problem is of high relevance and is likely to have a considerable impact in the chemistry community, I see myself using it, in particular if it is paired with SMILES. Nevertheless, my major concern is about the clarity of the paper, but I think that beyond clarity, the lack of the full or at least partial code should be available from the beginning for reviewers. This would make the evaluation of the article much easier.

##########################################################################Pros:

1. The paper addresses an interesting problem that, if solved, every chemist would like to make use of it in everyday life.

2. The proposed method uses and intermediate step in the learning process to encode a higher complexity as well as information. This provides flexibility that is reflected in a marginal gain in error prediction compared to other methods but it needs a lower number of data points.

##########################################################################
Cons:
Abstract: The abstract should be better structured. It should state first what the framework is or the context in which this work is set. The proceed to the specifics and the technical part.

A specific motivation for the use of the mediating representation is missing.

Be specific, “relatively low number of data points” doesn't provide any information.

The authors repeatedly use “strong labels are expensive”, just for the sake of completeness, would be good to be explicit in this fact instead of assuming that the reader will infer what the meaning is.

The authors state: “In order to measure empirically the performance of our method of self-labeling fully mediating representations we start with a pre-trained model and perform two steps.” What type of model or trained on what? It should be specific.

The reference “Slot attention” should be described more in detail and the advantages of the authors’ method over the Hungarian algorithm should be very clear.

I think a clear sentence is missing to describe in detail what the authors mean by strong and weak labels. In this regard, the section “Weak Supervision” is not straightforward to read. It should be rewritten or rearranged to make a better and clear reading.

“We also assume the map E(v) which gives all allowed graph edits for the graph v” it is not clear.

What is the origin on the such a different performance on the Indigo and Maybridge datasets.

Fig. 5 and 6 nicely summarise the good performance of the model, but in order to understand better the method and its limitations or type of graphs that struggle with, it would be good to present out-layers where the system doesn't work.

---

> ### Author Response · Authors · 2020-11-16
> **Response to AnonReviewer1**
>
> First of all your feedback and review is much appreciated. We are also very happy to make our code available for the community. Please have a look here: https://github.com/iclr2021-paper1739/workflow-paper1739 .
> Our plan is to make the complete version of the code available after the review process.
>
> To address your concerns we also submitted a new version of the manuscript with the new updates which are summarised below:
>
> * Regarding the abstract: Thanks for the suggestion we have updated the manuscript accordingly.
> * Regarding the specific motivation of the fully mediating layer: This mechanistic prior restricts the space of models to models that follow the mechanistic assumption. It is our hypothesis that this prior would enable to generalize better.  As a side effect this assumption also helps in making the model more interpretable. To showcase the interpretability we would also like to point to the notebook available in the now available github repo.
> * Regarding relative low datapoints: We use 4000 datapoints in comparison with over 57 Million datapoints used in  paper[Ref 30] to train their model.
> * Regarding expensive strong labels: Labeling manually 20 images took us about 2 hours. It is a manual process which requires intermediate organic chemistry knowledge. This in contrast with datasets with images together with SMILES which are more widely available.
> * Regarding pre-trained model: We pre-train a ChemGrapher [21] model wherefore, corresponding to the pipeline described in the work of Oldenhof et al. [21], we sample about 130K chemical compounds from ChEMBL in SMILES format and artificially generate, using an RDKit fork (https://github.com/biolearning-stadius/rdkit), a rich labeled dataset with 2D images of chemical compounds.
> * Regarding slot attention: the Hungarian algorithm is limited to only sets while in our case we need to map more complicated structures composed of different atoms connected with different bonds for which we need graph alignment.
> * Regarding strong and weak labels: Thank your for the feedback, we have revised the complete paragraph concerning weak supervision in our manuscript.
> * Regarding map E(v): Thank you for the feedback, we now explicitly introduce the concept of map E(v) in our manuscript.
> * Regarding difference in performance between Indigo and Maybridge dataset: Visually we can observe that the Maybridge dataset contains images where the style is closer related to the training images style used for the pre-trained model compared with the images in Indigo dataset where the style of images is quite different. Therefore we expect a significant worse starting performance of the pre-trained model on the Indigo dataset compared with the Maybridge dataset.
> * Regarding limitations of our method: There is need to have some minimum performance on the dataset in the new domain in terms of graph accuracy. Changing and tuning the map E(v) could help in this regard but is something which could be explored more in detail. Our plan is to add some examples where our method fails to the appendix before rebuttal deadline.
>
> Ref [30]: Joshua Staker, Kyle Marshall, Robert Abel, and Carolyn M. McQuaw. Molecular Structure Extraction from Documents Using Deep Learning. J. Chem. Inf. Model., 59(3):1017–1029, March 2019. ISSN 1549-9596. doi: 10.1021/acs.jcim.8b00669. URL https://doi.org/10.1021/acs.jcim.8b00669.
> [21]  Martijn Oldenhof, Adam Arany, Yves Moreau, and Jaak Simm.  Chemgrapher: Optical graphrecognition of chemical compounds by deep learning.arXiv preprint arXiv:2002.09914, 2020.doi: 10.1021/acs.jcim.0c00459.

---

### Official Review · AnonReviewer3 · 2020-10-29
**Self-Labeling of Fully Mediating Representations by Graph Alignment**

**Rating:** 5
**Confidence:** 4

**Review:**

The authors propose a domain adaption technique for self-labeling for strong/expensive planer graph labels given normal labels. For the application of molecular graphs, a graph alignment method on the planer graph level is proposed to find an isomorphism with a minimal edit distance of the predicted strong labels. The results show the proposed method can gradually correct the strong labels and improve the prediction performance with interpretable explanations.

However, there are several concerns about the paper:
1. The correction requires close edit distances between wrong labels and correct labels, and thus requires a relatively accurate U->V function. A plot regarding the correct percentage related to the accuracy of the initial U->V function will be interesting.
2. How are strong labels picked? The authors mention the strong labels are "a selection of these 4,000 datapoints". But ablation study on the selection, size, and quality(edit distance to other graphs) of those datapoints should be investigated.
3. The cost for the optimization problem argmin|e| is not discussed. The normal complexity for a single datapoint is N^d where N is the number of distance 1 editing and d is the distance allowed. A faster searching algorithm with domain knowledge is expected.
4. For the other methods mentioned in table 1, are strong labels required? The percentage of strong labels should be reported for a fair comparison. The performance with different initial correct percentage will be interesting to investigate.
5. The generalization of the model is not discussed. The adaption only works when the error of the prediction strong labels is smaller than k edit distance. However, the prediction depends on the distribution of the strong labels, the quality of the pre-trained models, the size of the graph, etc.

Other questions:
1. The structure of the paper can be improved. For example, the background of chemical structure recognition and the settings could be introduced at the beginning of the paper.
2. In the experiments part, why the percentage correct decrease with iterations?
3. In the graph alignment part, why to introduce sub-graph isomorphism?
4. What is the relationship between function U->V, V->W, the segmentation network, the classification network?
5. The segmentation network uses 134K  images. How is this data related to the training data?

Overall, the idea of gradually correcting strong labels using graph alignment is interesting, but more discussions and results are required to make the paper stronger.

---

> ### Author Response · Authors · 2020-11-18
> **Response to AnonReviewer3**
>
> We very much appreciate given feedback. We would like to give answers to raised concerns and questions.
>
> Concerns:
>
> 1. In the case the final graph (W) is correct we assume that it is highly likely that also the planar embedding (V) is correct. In this case, the accuracy of initial U->V is more or less equivalent to the iteration 0 model (where no domain adaptation had taken place). We have made this clear in text and also clarified the exact accuracies.
> U->V function does not have to be accurate for all graphs. In our case studies, Maybridge and Indigo, we had very different starting accuracies, about 70% and 10%, respectively (U->W). The method might even work if the initial accuracy is 0% as the method can correct graphs with few mistakes using graph alignment.
> 2. The phrase is indeed ambiguous and we correct it in the text. To clarify, the method does not have access to any strong labels from the target domain (as a side note, we have now replaced the term strong label with rich label to avoid confusion with weak supervision methods). The main idea of the method is to generate these strong labels (V) using the target domain normal labels (W). The generation process only uses strong labels (V) that have a very high chance of being correct because the generated strong label is
> (1) the predicted V itself (when the graph alignment gives 0 error match), or
> (2) the corrected version of V by the graph alignment (when the graphs do fully align).
> 3. N=number of atoms in the predicted molecule
>     d=number of allowed substitutions
>   a=number of possible atom types (e.g. O, C, N)
> Order of complexity for naive full search, which we used, is $O((N*a)^d)$ because to obtain high quality rich labels (V) we limited d to 2. To improve the speed a backtracking approach could be applied that avoids scanning the whole space of edits and backtracks when the alignment is not possible with the given remaining budget of edits.
> 4. Our plan is to make this table more clear and update it in the manuscript before the rebuttal deadline.
> 5. As discussed before it only really breaks down if all of the target domain samples have error larger than k. Even in the case if there are no samples with small enough error, one could try increasing k and obtaining noisy strong (rich) labels, which might bootstrap the domain adaptation, so that in the 2nd iteration there are some samples with error within k. We leave this option as a future research.
>
> Other Questions:
>
> 1. Thank you for the suggestion, we plan to update the manuscript.
> 2. The drop in performance after some iteration(s) was only observed in the case where a naive strategy of upsampling (upsample to a fixed number of target domain samples) was used. While if we take into account the rare atoms when upsampling, so that rare atoms are upsample relatively more, we did not observe this problem. Our hypothesis is that upsampling naively could in some cases cause a drop in performance for the rare atoms/bonds.
> 3. agreed, we dropped it.
> 4. For U->V we need both segmenation network and classification networks. For V->W we don’t need a network we can use a fixed projection that forgets the planar embedding.
> 5. So the 134K images used for the segmentation network are actually about 114K images from source domain also used to pretrain ChemGrapher114K augmented with about 20K rich-labeled images (after using our method and upsampled) from target domain.

---

### Official Review · AnonReviewer4 · 2020-10-30
**Recommendation to reject**

**Rating:** 4
**Confidence:** 3

**Review:**

##########################################################################

Summary:

The paper describes a method to convert 2D molecular images to molecular graph structures, with applications in extracting raw chemical structures from journal articles and other publications. The model has two components: a semantic segmentation network that first predicts the location of the atoms and bonds in the image, and a series of classification networks that classifies each segment. The paper proposes some domain adaptation techniques to reduce the amount of expensive ‘pixel-wise’ labels required for training the segmentation network

##########################################################################

Reasons for score:

Overall, I currently vote for rejection. I have some questions about the current evaluation setup that I hope the authors could clarify

##########################################################################

Strengths:

*The proposed iterative strong labeling and graph alignment framework seems to improve the performance of the pre-trained model

Weaknesses:

*The model evaluation seems to show some mixed results. On the Mayfield dataset, there is an improvement over some baseline models, but on the Indigo dataset, the proposed model seems to be perform significantly worse than the model described in the citing reference 30. (~40% vs ~80%)

*I found that some parts of the paper were difficult for me to understand (see below)

##########################################################################

Questions and other comments:

*Paper clarity:
**I think there should be more information on how the underlying Chemgrapher model converts 2D molecular images to the molecular graph structures. How does the model actually construct the graph structure? Also, there is a lot of analysis on the image segmentation part of the model (eg figure 5, 6). Does the image classification part of the Chemgrapher model play any significant role?
**Information about the Indigo and Maybridge dataset could be provided in a more accessible way. Eg the total number of examples in each dataset
**Some additional information about how the original model was pre-trained would be useful

*I think there needs to be more justification for why this proposed approach [option 1] of mapping the 2D molecule image to the molecular graph structure via an intermediate representation that explicitly identifies all the atoms and bonds in the image is preferred over the alternative approach [option 2] of directly mapping the 2D molecule image to the molecular graph structure (eg using by outputting a smiles text representation that can be converted to the molecular graph). The cost of option 1 is that it requires the very expensive pixel-wise labels that describes the locations of atoms and bonds in the 2D molecule image to train the segmentation model, thus motivating the domain adaptation part of this work. In terms of pure performance, it’s not clear to me that option 1 is superior, for example, [ref 30] which directly predicts the molecular smiles from the 2D molecular image [option 2] can attain ~80% in the indigo dataset, while this proposed approach seems to attain only ~40%

*Figures 2 and 3 show various performance metrics over multiple iterations of the re-training. How did you decide which iteration to stop the re-training? Also, how computationally expensive is it to perform the iterative re-training procedure?

*In Figure 3b, the performance at iteration 0 is ~72%, which I’m understanding to be the vanilla performance of the Chemgrapher model on the Maybridge dataset? But Table 1 shows that the performance of the Chemgrapher model on Maybridge is 83.3%

*What is the reasoning for allowing a max of 2 node substitutions or 1 edge substitution for the ‘correcting graph alignment’ case?

*In the future, it would be interesting to see how this proposed method compares in this recently published benchmark: https://github.com/Kohulan/OCSR_Review. NB: out of scope for this ICLR submission since it was published after the paper submission deadline


Ref [30]: Joshua Staker, Kyle Marshall, Robert Abel, and Carolyn M. McQuaw. Molecular Structure
Extraction from Documents Using Deep Learning. J. Chem. Inf. Model., 59(3):1017–1029,
March 2019. ISSN 1549-9596. doi: 10.1021/acs.jcim.8b00669. URL https://doi.org/10.1021/acs.jcim.8b00669.

---

> ### Author Response · Authors · 2020-11-20
> **Response to AnonReviewer4**
>
> Thank you very much for your effort in reviewing our work. We would like to give answers to your questions:
>
> * paper clarity:
> ** After the first step of the ChemGrapher model, the segmentation network, the predicted segments are processed so that for every segment the center of mass is calculated. These centers of mass would be the atom/bond/charge candidates to be classified by the classification networks. We also made a notebook available on https://github.com/iclr2021-paper1739/workflow-paper1739  where this whole process is implemented in the ‘predict’ step using the ChemGrapher model.
> 	So we could say that both steps are important/significant in ChemGrapher. However, the segmentation step is maybe performing the ‘heavy lifting’ of the model so that the classification networks have a relatively easy task.
> 	We have updated the manuscript accordingly.
> ** Thank you for the suggestion we provide now a summary table about 2 datasets in the manuscript.
> ** Thanks, we added this to the manuscript. More specific:
> 	We pre-train a ChemGrapher [21] model wherefore, corresponding to the pipeline described in the work of Oldenhof et al. [21], we sample about 130K chemical compounds from ChEMBL in SMILES format and artificially generate, using an RDKit fork, a rich labeled dataset with 2D images of chemical compounds.
> 	We updated the manuscript accordingly.
>
>
> * The model of [ref 30] uses a Indigo training dataset of 54 million datapoints while ChemGrapher only uses about 130K datapoints from another domain and 4K (upsampled to about 20K) from the Indigo target domain to train. This gives a performance of 40% and is indeed lower than the 80% performance on Indigo dataset from [ref 30]. However ChemGrapher was trained with 3-4 orders of magnitude less data from the Indigo dataset which makes the result of 40% rather impressive although lower than 80%.
> The purpose of the Indigo domain was not to show superiority of the ChemGrapher approach above the other method [ref 30]  but to experiment if our method works on a target domain where the starting performance is rather poor (around 10%).
>
> * The aim of our experiments was to evaluate two main research questions:
> 1. What is the effect of the pre-trained starting performance on our method. Will our method work in a poor starting performance vs relative good starting performance?
> 2. What is the effect of exact graph alignment vs correcting graph alignment? Has correcting graph alignment a better effect on the performance  compared to exact graph alignment?
>
> These 2 main research questions could be confirmed with our experiments.
> However to reach a higher performance on the target domain more iterations could be considered together with using more datapoints from target domain.
> For the cost of retraining we refer to the appendix where we will add before rebuttal deadline the details about the computational costs on each training iteration.
>
> * Iteration 0 is indeed always the performance of the pre-trained (on source domain) ChemGrapher model. The performance metric of 83.3% is actually the best performance that was reported in Oldenhof et al on the Maybridge dataset. For this performance they added some manually rich labeled difficult hand picked datapoints from the target domain Maybridge dataset. Our plan is to update this table before rebuttal deadline so that the numbers are more clear.
>
> * The reasoning behind this is that in the case of a very constrained correcting graph alignment (max 2 node/ max 1 bond subst) it is highly likely to obtain high quality rich labeled datapoints. Also, as our method is an iterative method, datapoints that could not be graph aligned in a previous iteration can still be graph aligned in a next iteration considering that a new graph prediction could be now closer to the true graph.
>
> * Thanks for the suggestion. We have carefully reviewed the benchmark and will consider to add also the tool Molvec in our experiments for the benchmarks of the Maybridge dataset. Our plan is to  add the results before rebuttal deadline.
> For the validation of our method we consider that the results on the Indigo and Maybridge datasets clearly show how the method is able to perform in domain adaptation having different starting performance.
>
> Ref [30]: Joshua Staker, Kyle Marshall, Robert Abel, and Carolyn M. McQuaw. Molecular Structure Extraction from Documents Using Deep Learning. J. Chem. Inf. Model., 59(3):1017–1029, March 2019. ISSN 1549-9596. doi: 10.1021/acs.jcim.8b00669. URL https://doi.org/10.1021/acs.jcim.8b00669.
>
> [21] Martijn Oldenhof, Adam Arany, Yves Moreau, and Jaak Simm. Chemgrapher: Optical graphrecognition of chemical compounds by deep learning.arXiv preprint arXiv:2002.09914, 2020.doi: 10.1021/acs.jcim.0c00459.

---

### Comment · Area_Chair1 · 2020-11-20
**Please, read rebuttals and start discussion if needed**

Dear Reviewers and Authors,
Thanks for starting the discussion.

Reviewers: please, check the rebuttals provided by the authors, verify if they replied properly and you are satisfied.
Possibly, give further feedback or make questions, only if needed and important for your final evaluation.
Please, be accurate and precise in your further requests, so that authors can understand and reply properly and focused.
Eventually, you need to revise your review and report final comments and rating.

Authors: please, check if there are further clarifications needed by the Reviewers.
Please, be focused in your final answers and avoid to ask questions to Reviewers, if not absolutely necessary.

For All: please, I would avoid a long chat-like discussion, a couple of iterations are affordable on a few specific points to be clarified, but no more.

Thanks and best regards

AC

---

### Author Response · Authors · 2020-11-24
**Final rebuttal revision available**

Dear Reviewers, Area Chair,

We have made our final rebuttal revision now available which should address the comments that were still pending during discussion phase.
Reviewers: We appreciate all of your valuable feedback which enabled us to make the paper stronger.

Thanks again and best regards,

ICLR 2021 Conference Paper1739 Authors

---

### Decision · Program_Chairs · 2021-01-07
**Final Decision**

**Decision:**

Reject

**Comment:**

The paper proposes a graph aligning approach generating rich and detailed labels given normal labels. Authors cast the problem in a domain adaptation setting, considering a source domain where "expensive" labels are available, and a target domain where only normal labels are available. The application scenario is the prediction of chemical compound graphs from 2D images, where a fully mediating layer is introduced to represent using a planar embedding of the chemical graph structure to be predicted.

The paper received ratings all below-threshold.
The main issue transversal to all reviewers relate to clarity of the presentation.
Clear motivations for some of the adopted choices of the method and of the experimental procedure were also missing. In particular, missed to provide the clear usefulness of the main paper's contribution, i.e., to neatly show the importance of the mediating layer (ref. R4, R2).

The lack of important details in the method description and experimental results were also deemed a major shortcoming: cost of the optimization, model generalization not discussed, contradictory results on the different datasets considered, comparative analysis, partial ablation, are among the main quoted remarks.

Authors' rebuttal is carefully provided in general, but several issues are still remaining.

Hence, overall, given the above issues, I consider the paper not yet ready for publication in ICLR 2021.